# Identifying Important Factors for Depressive Symptom Dynamics in Chinese Middle-Aged and Older Adults Using a Multi-State Transition Model with Feature Selection

**DOI:** 10.3390/bs15111501

**Published:** 2025-11-05

**Authors:** Chuoxin Ma, Tianyi Lu, Yu Li, Shanquan Chen

**Affiliations:** 1Guangdong Provincial Key Laboratory of Interdisciplinary Research and Application for Data Science, Beijing Normal-Hong Kong Baptist University, Zhuhai 519088, China; u430202703@mail.uic.edu.cn (T.L.); yuli@uic.edu.cn (Y.L.); 2Department of Population Health, Faculty of Epidemiology and Population Health, London School of Hygiene and Tropical Medicine, London WC1E 7HT, UK

**Keywords:** depressive symptoms, symptom conversion pattern, aging population, multi-state Markov model, feature selection

## Abstract

Depressive symptoms are increasingly common in middle-aged and older adults and have become a major public health problem. People may experience transitions across different underlying states due to symptom variability over a course of many years. And risk factors may have different impact on different symptom states. However, existing research rarely considers the identification of important factors related to symptom conversion. The purpose of this study was to examine the risk associated with transitioning between various stages of depressive symptoms and their influencing factors, utilizing a multi-state model with a simultaneous feature selection method. We used the four waves of data from the China Health and Retirement Longitudinal Study (CHARLS) and 3916 participants were selected after screening. Five states of depressive symptoms were defined including no symptom, new symptom episode, symptom persistence, remission and relapse. We included 13 variables on demographic background, health status and functioning, and family and social connectivity, along with their interactions. Multi-state models were used to evaluate the risks of state transitions. The regularized (adaptive Lasso) partial likelihood approach was employed to simultaneously identify the important risk factors, estimate their impact on the state transition rates and determine their statistical significance. There were 1392 new depressive episodes events, 402 symptom persistence events, 639 remission events and 118 relapse events. We identified nine significant risk factors for the new onset of depressive symptoms: urban–rural residence, sex, retirement status, income, body pain, difficulty with basic daily activities, social engagement, education by income interaction and number of conditions by income interaction. The effects of the identified risk factors on new symptom episode weakened as those symptoms became persistent or went into remission. In terms of symptom relapse, sex by age was identified as a significant influencing factor. This study identified key factors and explored their effects on the various depressive symptom states among older Chinese adults. The findings could serve as a foundation for the development and implementation of targeted policies aimed at enhancing the mental well-being of China’s elderly population.

## 1. Introduction

Elevated depressive symptoms are a common public health concern globally. According to a systematic review ([37]), the global prevalence of elevated self-reported depressive symptoms over the period 2001–2020 was 34%. Depressive symptoms can have a wide range of negative impacts on a person’s life, affecting both physical and mental health. Depressive symptoms are associated with a higher risk of chronic diseases and multimorbidity such as heart disease, digestive disease, liver disease, kidney disease, lung disease, and memory-related conditions ([44]). Depressive symptoms account for 7.8% of total personal expected medical spending in China, reflecting the significant impact of mental health conditions on individual healthcare expenditure ([18]). Given that there is a high prevalence of depressive symptoms among Chinese adults aged 45 years and older ([47]), it leads to severe public health resource burden and poses a significant economic strain. Preventing and managing depressive symptoms can help reducing the economic burden, and effective interventions rely on the identification of factors related to the development of depressive symptoms in different stages.

In recent years, there have been many works investigating depressive symptoms based on longitudinal survey data and that have shown that depressive symptoms evolve dynamically over time, especially in the elderly population ([21]; [43]). Previous studies discovered that elderly individuals who encountered their first episode of depressive symptoms stand a relatively high likelihood of enduring persistent or recurring episodes later in life ([16]; [27]). A multi-state model encompassing the states of no symptom, new episode and symptom persistence showed that loneliness was the most relevant factor in predicting symptom episode emergence and persistence in old-age ([10]). Other multi-state analysis of the course of late-life depressive symptoms found that somatic disease increased the risk of progressing from no symptoms to subsyndromal symptoms without depression, while decreasing the recovery rate. Physical activity and a richer social network increased the likelihood of recovering ([41]). An analysis of depressive and anxiety symptom transitions in coronary heart disease patients revealed that body mass index, education, and physical activity were linked to the progression from no symptoms to mild symptoms and from no symptoms to moderate/severe symptoms in coronary heart disease patients ([29]). Other researchers investigating the association between chronic disease status and depressive symptom transitions found that both single diseases and multimorbidity were significantly associated with an increased risk of transitioning from no symptoms to mild depressive symptoms. Additionally, single diseases were significantly associated with a reduced risk of transitioning from mild to severe depressive symptoms ([46]). Beyond examining the dynamics of depressive symptoms through state transitions, researchers also investigated the temporal pattern of depressive symptoms from the perspective of symptom trajectory analysis. Numerous studies have identified heterogeneity in the changing pattern of depressive symptom trajectory among the older adults ([3]; [5]; [9]; [49]). The existence of these distinct trajectory classes suggests that there are several underlying stages of depressive symptoms over the course of time.

To develop effective interventions for managing depressive symptoms, it is crucial to identify risk factors that influence each phase of the progression of depressive symptoms. Much research has been conducted to detect demographic and health-related factors that have a substantial impact on the onset of depressive symptoms. Lin et al. used the long short-term memory network approach to predict the onset of depressive symptoms in home-based older adults. Factors such as self-reported health status, cognition and activities of daily living disorder were identified as the top important variables ([25]). A recent study analyzed the influence factors associated with the incidence of depressive symptoms in middle-aged and elderly Chinese patients with diabetes using binary logistic regression. The presence of depressive symptoms was found to be influenced by various factors including age, education level, marital satisfaction, physical disability, and the presence of comorbid chronic diseases ([1]). Shen et al. investigated the risk of depressive symptoms in Chinese middle-aged and older adults with physical disabilities. LASSO regression analysis combined with binary logistic regression analysis was performed, and their results showed that living in a rural area, difficulties with activities of daily living and being female are important factors for the presence of depressive symptoms ([36]). Another study focused on depressive symptoms among older Chinese adults with chronic diseases, identifying key predictors of depressive symptoms using machine learning algorithms. Factors such as health satisfaction, sleep duration, life satisfaction, number of chronic diseases, and education were found to be more significant in males than in females ([53]). In addition to these identified factors, the age of of symptom onset, even in the absence of a formal clinical depression diagnosis, is also an important factor that is worthy of investigation. Based on the Center for Epidemiologic Studies Depression scale, trajectories of depressive symptoms were modeled and it was found that depressive symptoms were highest in young adulthood, decreased throughout midlife, and increased again in later life ([39]). Liang et al. also showed that the trajectory of depressive symptoms varies with age, manifesting in distinct patterns such as persistently high, increasing, and decreasing levels ([24]).

These studies provided insights for understanding the risk factors associated with depressive symptoms. However, most of them only considered a single static state in the course of symptom evolvement and failed to reveal the temporal dynamics. The longitudinal analysis of symptom trajectories can provide insights into symptom evolution. However, there has been limited work considering the shifts from one trajectory class to another over time. Current methods of trajectory analysis are insufficient for modeling the risk of transition between various trajectory membership classes at a particular time. This aspect is crucial for analyzing the conversion between the underlying stages of depressive symptoms. The multi-state modeling approach enhances our understanding of the progression of depressive symptoms by effectively modeling transitions between various stages and evaluating risks at any given time point. Therefore, we aim to use multi-state models to explore the temporal dynamics of depressive symptoms. The previous multi-state studies referenced above conceptualize symptom states solely based on severity level (no symptom, mild symptom and severe symptom) ([10]; [29]; [41]; [46]). These models did not capture trajectories of relapse or remission, failing to differentiate between individuals who recovered and then relapsed, and those who had been symptom-free before encountering new-onset symptoms. They potentially overlooked important patterns in the long-term course of depressive symptoms. Moreover, previous multi-state studies typically used the same set of risk factors for all state transitions. It is highly probable that each depressive symptom state has its own key risk factors. And the underlying mechanism may, in fact, be state transitions, each governed by a unique set of covariates in their respective transition-specific models. Yet, in many cohort studies, a large number of risk factors are measured in advance and it is hard to determine which ones should be included in the analysis. One might choose a large set of covariates with the expectation that all potential risk factors are included, and apply the same set of covariates to all state transitions. However, many factors may have no impact on certain symptom stages, and implementing a multi-state analysis that incorporates the same extensive set of covariates in each state transition-specific model could lead to a large number of parameters to estimate, resulting in overfitting or inefficiency. Conversely, if we only include a small set of covariates in each state-transition model to avoid overfitting, there is a risk of neglecting some important risk factors for some specific states. This could lead to substantial residual dependence in the modeling of state transitions, resulting in invalid inference ([12]).

To address these issues, we utilized a multi-state modeling approach that incorporated a more detailed classification of depressive states, reflecting specific symptom phases over time: no symptoms, new onset, persistence, remission, and relapse. This approach allowed us to examine the complex dynamics of depressive symptom progression among middle-aged and elderly Chinese adults. Alongside this, we utilized feature selection methods to identify the key risk factors associated with each symptom stage. This dual-method strategy provided a more comprehensive and nuanced understanding of the factors influencing different state transitions. To the best of our knowledge, this is the first study that integrates feature selection with multi-state models for the analysis of depressive symptoms. We believe that by uncovering the dynamics of the course of depressive symptoms among the older Chinese population, it contributes to a deeper understanding of mental health in this demographic, which is often overlooked in global mental health discourse. Identifying the crucial risk factors can inform targeted interventions and preventive measures, potentially reducing the societal burden of managing depressive symptoms.

## 2. Methods

### 2.1. Study Participants

This study used data from the China Health and Retirement Longitudinal Study (CHARLS) ([51], [52]), which is a survey based on a sample of households with members aged 45 years or above. The baseline national wave of CHARLS was initiated in 2011 and included interviews with 17,708 individuals in 150 counties/districts and 450 villages/resident committees. Then the participants were followed up every two years in the subsequent 2013, 2015, 2018 and 2020 national waves. Data were collected based on the Health and Retirement Study and related aging survey questionnaires including the following modules: demographics, family structure/transfer, health status and functioning, biomarkers, health care and insurance, work, retirement and pension, income and consumption, assets (individual and household), and community level information. It provides a wide range of information from socioeconomic status to health conditions over a long time span with about 67% of the participants being followed for more than 7 years.

This study utilized the first four national waves of CHARLS data. We did not include the 2020 wave because the impact of COVID-19 on depressive symptoms could introduce potential bias. Our primary focus is on depressive symptoms during normal times, prior to the pandemic. The primary inclusion criterion for this study was individuals with complete scores on the Center for Epidemiologic Studies Depression Scale (CES-D) across all four waves. Participants with three or more missing items on the CES-D form in any wave were considered to have missing data. Further exclusion criteria of this study were as follows: (1) individuals aged below 45; (2) those who showed depressive symptoms at baseline or reported receiving any form of treatment for emotional or psychiatric problems—including psychiatric/psychological therapy, antidepressants, tranquilizers, sleeping pills or other medications—in any of the four waves; (3) individuals with a high rate (more than 30%) of missing data for risk factors, where imputation would be inaccurate. Based on these sample selection criteria, 3916 subjects were included for the final data analysis. Figure 1 shows a flow diagram of the study sample selection process.

### 2.2. Assessment of Depressive States

Depressive symptoms and their underlying states were defined according to a severity score calculated based on the 10-item CES-D form. The CES-D has established validity and reliability in detecting both clinical and non-clinical depressive symptoms and has been widely used to assess depressive symptoms in community and population-based epidemiological studies. Numerous studies have validated the CES-D as a reliable measure for assessing depressive symptomatology in older adult populations ([2]; [6]; [31]). During the follow-up process of CHARLS, participants were asked the frequency of experiencing the following items to evaluate how they felt and behaved over the previous week: (1) I was bothered by things that don’t usually bother me; (2) I had trouble keeping my mind on what I was doing; (3) I felt depressed; (4) I felt everything I did was an effort; (5) I felt hopeful about the future; (6) I felt fearful; (7) My sleep was restless; (8) I was happy; (9) I felt lonely; (10) I could not get “going.” Each item was measured using a 4-point Likert scale from 0 (rarely or none of the time) to 3 (most or all of the time). Items 5 and 8 were reversed-scored. The total score ranges from 0 to 30 with higher scores indicating greater severity. We removed subjects with three or more missing items on the CES-D form ([34]) and defined a depressive symptom episode as observing a total CES-D score equal to or higher than 12 ([4]; [44]; [54]). Symptom persistence was defined as observing a score maintaining at 12 or above in a consecutive follow-up visit, after the new depressive symptom episode. Remission was defined as observing a score reduced to 12 or below in a subsequent follow-up visit, after the occurrence of new depressive symptom episode. Symptom relapse was defined as another depressive symptom episode after remission. While both the symptom persistence and relapse states involve the presence of significant depressive symptoms, they are operationally defined to represent distinct phases in the dynamic course of depressive symptoms. The key distinction is that individuals in the persistence state have not experienced remission ([10]; [16]), whereas those in the relapse state have transitioned back to elevated symptoms after remission ([11]; [27]). By modeling these as separate states, our analysis can investigate whether the risk factors for recurrent depressive symptoms are different from the risk factors for remaining stuck in a chronic depressive state. Since our inclusion criterion required complete CES-D scores across all four waves, individuals who died before completing all four waves of follow-up were not included in our sample. Consequently, death as a competing risk event is not considered in this study. Similar analysis criteria can be found in ([46]). There were 1392 new depressive episodes events, 402 symptom persistence events, 639 remission events and 118 relapse events observed in the sample data.

### 2.3. Measurement of Covariates

The risk factors considered include residence, sex, age, highest level of education attained (education), marital status, processed retirement or not (retired), household per capita income (income), number of diagnosed chronic conditions (number of conditions), currently feel any body pain or not (pain), have any difficulty with basic activity of daily living (BADL disability), attain any of the social activities (social engagement), contact with children (child interaction), and any transfer from non-coresident children (child support).

The range of the missing rate of these risk factors was between 0% and 4.4%. We used the multivariate imputation by chained equation method ([42]) to impute the missing data. Outliers were detected using the median absolute deviation approach. The characteristics of the study subjects are summarized in Table 1. The quantitative variables including age, income and number of conditions were normalized to fall within the range of 0 to 1 before model fitting and variable selection.

In addition to independent effects, we investigated potential interactions among the 13 variables. Given the large number of possible interaction terms, we employed a two-stage selection process: initial feature screening to identify promising interactions, followed by variable selection within the multi-state model. We employed an adaptive model-free feature screening method ([7]) to identify potentially important interactions. The following interaction terms were selected as candidates for further evaluation and underwent feature selection within the multi-state model: education × income, number of conditions × income, number of conditions × sex, sex × age, and age × education.

### 2.4. Multi-State Model with Feature Selection

Multi-state models are a type of statistical model used primarily in survival analysis, and they are particularly useful when individuals can transition between different states over time ([23]). The states could represent different stages of a disease process or different levels of a certain behavior. The individual transitions from one state to another over time, and the goal of the analysis is to understand the rates of these transitions and the associated risk factors. In this study, we defined five states: no symptom, new depressive symptom episode, depressive symptom remission, symptom persistence and relapse. Let 
S={hg:12,23,24,34,45}
 represent the set of all possible transition paths and they are presented in Figure 2. We used the following multi-state Cox model to evaluate the impact of risk factors on depressive symptom progression
(1)
$$\lambda_{i,hg}\left(t\right)=\lambda_{0,hg}\left(t\right)\exp\left(\beta_{hg}X_{i}\right),$$

where 
$\lambda_{i,hg}\left(t\right)$
 is the hazard rate at which individual *i* transitions from state *h* to state *g* at time *t*, 
$\lambda_{0,hg}\left(t\right)$
 is the corresponding baseline hazard, and 
$\beta_{hg}$
 is the unknown coefficient that describes the association between factors and the risk of transition between state *h* and *g*. The maximum L1-regularized (adaptive Lasso) partial likelihood approach ([28]) was used to simultaneously identify the important risk factors, estimate their impact on the state transition rates and determine their statistical significance. The coefficients were estimated as follows.
$$\begin{array}{cc} \hat{\beta }= & \underset{\beta }{\arg\min}\{-\frac{1}{n}\sum_{hg\in S}\sum_{i=1}^{n}\left(\beta_{hg}^{T}X_{i}-\log\left(\sum_{j\in R_{h}\left(t_{i,hg}\right)}\exp\left(\beta_{hg}^{T}X_{j}\right)\right)\right)+\\ & \sum_{hg\in S}\sum_{r=1}^{p}\frac{\lambda_{hg}}{\left|{\tilde{\beta }}_{hg}^{r}\right|}\left|\beta_{hg}^{r}\right|\}, \end{array}$$

where 
$t_{i,hg}$
 is the time point at which individual *i* moves from state *h* to state *g*, 
$R_{h}$
 is the at risk indicator, and 
$\tilde{\beta }$
 is the initial estimate obtained using standard partial likelihood method. All statistical analyses were carried out using R software.

## 3. Results

We used transition rate ratios (TRRs) along with their 95% confidence intervals to measure the relative risk of transition between different depressive symptom states. For binary data risk factor, the TRR is the relative risk of state transition for one group (when the binary variable takes value one) compared to the reference group (when the binary variable takes value zero), adjusting for all the other risk factors. For continuous data, the TRR is the relative risk for one unit of change in the variable of interest, adjusting for all the other variables. The estimated coefficients and their standard errors, as well as the TRR and their 95% confidence interval limits (CL) for the selected relevant features were calculated and reported in Table 2. The coefficients of all the other irrelevant risk factors were compressed to be zero. In the last column of the table, significant factors were marked with 
**
 for *p*-values lower than 0.001 and * for *p*-values lower than 0.05. We additionally plot the TRR of all statistically significant features (significance level chosen as 0.05) in Figure 3.

For females, currently feeling body pain and having difficulty with basic activity of daily living were significantly and positively associated with the risk of developing new symptom episode. Residing in urban areas, being retired, higher household income, and social engagement were associated with a significantly lower risk of a new symptom episode. As for the interaction terms, the positive effect of the number of conditions × income interaction suggests that the protective effect of income weakens as the number of chronic conditions increases. To be more specific, for individuals with a low number of chronic conditions, the protective effect of income dominates. As the number of chronic conditions increases, its effect attenuates the main effect of income. The negative effect of the education × income interaction suggests that the combined socio-economic advantage of higher education level and higher income was associated with a reduction in the risk of symptom onset. We also found that older females were more prone to symptom relapse based on the estimated effect of the sex × age interaction term.

We further conducted an analysis to explore the potential role of age of symptom onset across the four symptom states: new symptom episode, symptom persistence, symptom remission and symptom relapse. Given that age-of-onset is undefined for participants who never exhibited significant symptoms during the study (individuals who were censored), it could not be incorporated as a baseline predictor in the multi-state model alongside other risk factors. We therefore conducted a descriptive analysis, examining its distribution for subjects who have experienced depressive symptoms during the study period. Visual comparison of the histograms (Figure 4) revealed that all distributions were slightly right-skewed, with a common peak occurring between 55–65 years of age. Using a threshold of 60 years ([8]; [14]; [17]; [35]) to distinguish early- from late-onset symptoms, we observed a higher frequency of early-onset cases among individuals in the persistent and relapse states. This suggests that an earlier onset is associated with a higher risk of symptom chronicity or recurrence.

## 4. Discussion

Investigating different depressive symptom states and their conversion patterns offers insights into the temporal progression of depressive symptoms. This understanding can facilitate early identification and intervention, potentially averting the development of severe depressive symptoms. In addition, identifying the most important factors of different state transition paths can help optimize the allocation of public resources, targeting those most at risk and devising personalized interventions. Such tailored strategies could substantially reduce the social and economic implications linked to depressive symptoms. To achieve these goals, this study explored a range of factors including demographic, health-related, and family social elements, and their effect on the conversion of depressive symptom states in middle-aged and elderly Chinese residents.

This study contained two strengths: (1) Simultaneous analyses of the various stages of depressive symptom progression, including the new onset of a symptom episode, symptom persistence, remission, and relapse were provided. In contrast to conventional methods such as the standard Cox model and trajectory analysis of longitudinal data, we concurrently analyzed multiple depressive symptom pathways. The multi-state modeling approach offered valuable insights into the complex evolution of depressive symptoms over time. Our study revealed novel findings such as, during the transition from symptom onset to persistence, the impact of the associated risk factors declined. Conversely, the impact of the sex-by-age interaction amplified as an individual progressed from symptom onset to symptom relapse. (2) We implemented, for the first time, feature selection alongside multi-state models to determine the key risk factors. Our analysis uncovered that the set of important factors changed along the progression of depressive symptoms.

For the new onset of depressive symptoms, factors such as urban–rural residence, sex, retirement status, income, body pain, difficulty with basic daily activities, social engagement, education by income interaction and number of conditions by income interaction were identified as important risk factors. These findings align with previous studies and underscore the importance of targeted intervention policies. Being female and having greater number of chronic conditions are positively associated with the risk of developing depressive symptoms. Having an education level of primary school or above, along with residing in urban areas are negatively associated with the risk of new depressive symptom episode in Chinese middle-aged and older adults ([48]). In addition, participants with higher household income and less BADL disability had a significantly lower risk of developing incident depressive symptom ([54]).

Living in urban areas in China is associated with a lower risk of depressive symptoms for several reasons: The rapid urbanization of China represents upward social mobility. This transition is typically associated with better educational opportunities and greater economic resources. On the other hand, rural areas tend to have a higher proportion of elderly people, often "left behind" by their adult children who migrated to cities ([38]). This situation can lead to feelings of isolation and loneliness, which can contribute to a higher risk of depressive symptoms ([19]). Rural elderly residents also have poorer access to public transportation, subsidized housing, high quality medical care, mental health services and pensions ([45]).

Gender is associated with the risk of depressive symptoms due to several factors. Societal and cultural factors often impose multiple roles and responsibilities on women, such as childcare and caregiving, increasing their stress levels and, consequently, the risk of depressive symptoms. Women also might be more prone to internalizing negative thoughts and emotions, making them more vulnerable to depressive symptoms ([15]). Some studies also found that older women are more likely to experience low perceived financial comfort and health problems compared to older men, and these may be related to the fact that women experience more depressive symptoms than men ([13]).

Limitations in BADL are linked to depressive symptoms due to their impact on physical function and social activities. Impairments such as loss of lower body strength or reduced mobility can negatively affect an individual’s mood and limit their ability to engage in various activities, including social interactions. This can lead to feelings of isolation and ultimately, occurrence of depressive symptoms. Additionally, individuals with lower BADL levels, indicating greater impairment, may struggle with self-care, further contributing to depressive symptoms. Therefore, maintaining or improving BADL could potentially reduce the risk of depressive symptoms ([26]).

Chronic conditions are associated with depressive symptoms due to their impact on physical ability, independence, and social interactions. These conditions often lead to disability and reliance on others, reducing self-confidence and increasing feelings of loneliness. Additionally, the physical and psychological stress of adapting to these conditions can exacerbate depressive symptoms ([30]). Overlapping neuropathology mechanisms between pain and depressive symptoms, as well as self-perceived health-related quality of life may explain the identified link between body pain and depressive symptom onset ([33]).

Income is associated with the risk of depressive symptoms and a previous study found that the impact of poverty on depressive symptoms is mediated by the deterioration of household living conditions and a decrease in social participation ([20]). Furthermore, current mental health policies in China primarily address severe mental illnesses. This, coupled with an imbalance of mental health resources in poorer areas, means that low-income individuals often lack access to essential mental health services ([50]).

The connection between education and depressive symptoms can be elucidated from various perspectives. Education can increase individuals’ productivity and earning potential, leading to economic benefits across the lifespan, influence health outcomes by shaping individuals’ access to health-promoting resources, and contribute to cognitive and emotional resilience, which may protect against the onset and persistence of depressive symptoms ([32]). Education may also improve allocative ability, and hence generally higher education is associated with better health and lower prevalence of depressive symptoms ([22]).

Furthermore, we found that the impact of the above-mentioned risk factors weakened when an individual progressed to symptom persistence and symptom remission. This contributes new insights by suggesting that urban–rural residence, retirement status, education, income, body pain, difficulty with basic daily activities, number of conditions, social engagement and sex served as primary determinants of symptom onset, but their effects diminished over the dynamic course of symptom states, including both persistence and remission.

Lastly, sex and age interaction was found to be a key determinant of symptom relapse. This finding suggests that the course of depressive symptoms diverges across the lifespan for men and women in middle and older age. In addition, we discovered that the effects of sex and age intensified when an individual experienced a relapse, compared to when an individual encountered a new symptom episode. A prospective, observational study in Germany also found that women had a higher prevalence of recurrent depressive symptoms compared to men based on multiple logistic regression model ([40]). However this analysis did not consider the temporal aspects of symptom recurrence. The extension of time-to-event analysis to symptom relapse remains largely unexplored. To our knowledge, this is the first study to reveal that sex and age interaction played the pivotal role in determining the relapse of depressive symptoms, and the influence became more pronounced when an individual progressed from the state of symptom onset to symptom relapse.

These findings highlight the need for gender-specific mental health programs and integrated healthcare systems that combine mental health services with the management of chronic physical conditions. Enhancing access to mental health services and community support programs in rural areas may also help relieving the burden of depressive symptoms from a societal perspective. This could include increasing funding for rural mental health clinics, implementing mobile health clinics and telemedicine services, initiating community centers, recreational activities, and social programs. Additionally, it is important to provide more support for home care and rehabilitation services, and to establish community programs that provide social engagement and physical activities tailored for individuals with difficulty in BADL. Policy makers should also pay attention to education accessibility and ensure that children have access to at least primary education, regardless of their socioeconomic status. Adult education programs that allow individuals who did not complete primary school to gain basic literacy and numeracy skills could also be implemented. Policies that address Socioeconomic disparities are also essential as these are often intertwined with education levels and mental health outcomes.

This study is subject to a few limitations. First of all, the effects of the factors identified in this study do not establish a causal relationship. Future research could consider mediation analysis or causal analysis for multi-state transition models. Second, the assessment of depressive symptoms was conducted at fixed longitudinal survey time points. As a result, we can only approximate the timing of symptom state transitions to coincide with these scheduled visit times. It is likely that changes in depressive symptom status occurred prior to the longitudinal visits, which could lead to inaccuracies in the evaluation of the transition times. In such instances, the actual transition time point would be earlier than the scheduled survey time point, but the extent of this discrepancy remains unknown. Addressing this issue with more advanced methods such as multi-state analysis with interval censoring is a direction for future work. Third, a more nuanced perspective on symptom progression could potentially be achieved by further subdividing the persistence state into worsening and stable, and the remission state into partial and full recovery. However, such refinement was not feasible in the present study for two primary reasons: first, the lack of established clinical thresholds to reliably define these substates; and second, statistical power considerations, as further subdivision would fragment the sample size and compromise the stability of model estimates. Future research with larger longitudinal datasets and validated clinical benchmarks would be well-positioned to explore these symptom states.

In conclusion, our study demonstrates that different sets of risk factors are associated with different states of depressive symptoms. The multi-state models can well capture the temporal dynamics in the symptom state conversions. Coupled with the feature selection method, the most significant risk factors for new-onset of depressive symptom, symptom persistence, remission and relapse are identified. The findings of this study can guide policy-making and resource distribution within public health services.

## Figures and Tables

**Figure 1 behavsci-15-01501-f001:**
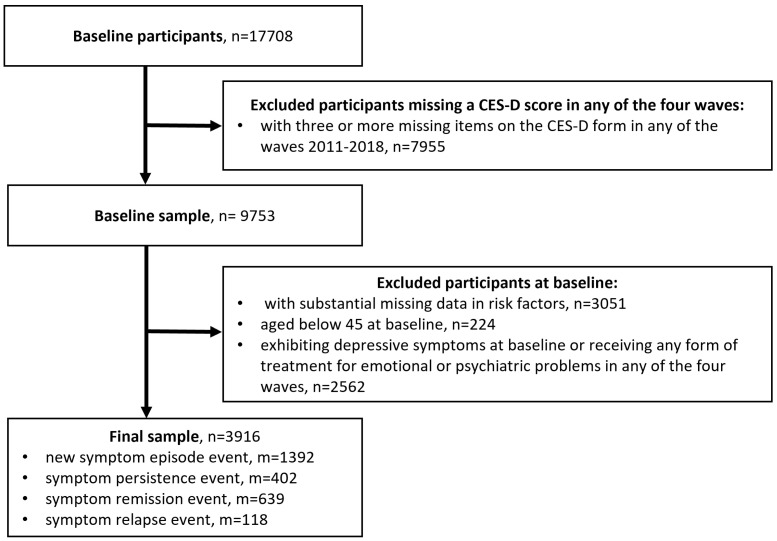
Study sample screening flowchart.

**Figure 2 behavsci-15-01501-f002:**
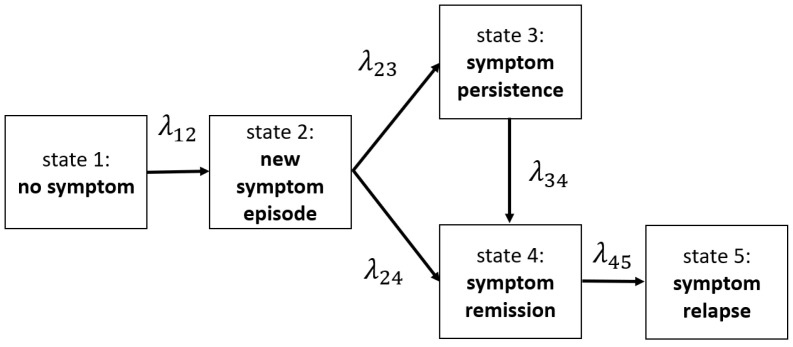
Illustration of the course of depressive symptoms with multi-state event diagram (
λ
 represents the risk of transition between different states).

**Figure 3 behavsci-15-01501-f003:**
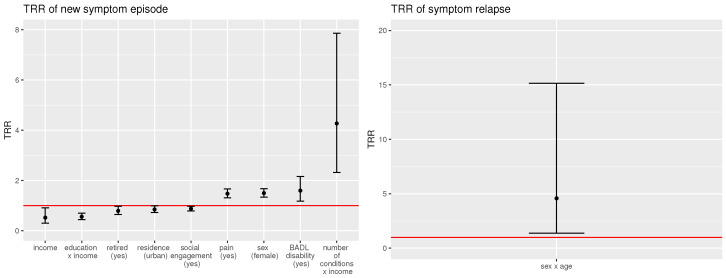
Statically significant risk factors and their TRR confidence intervals.

**Figure 4 behavsci-15-01501-f004:**
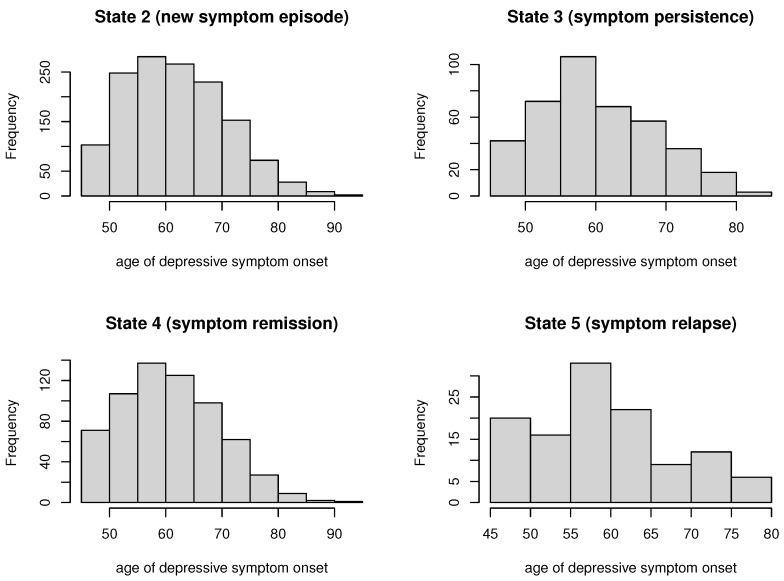
Distributions of age at depressive symptom onset across symptom states.

**Table 1 behavsci-15-01501-t001:** Variable description and study population characteristics.

Variable	Description	Mean/Prop	SD
**(1) Demographic background**			
residence	1: urban community, 0: rural village	21.87%	-
sex	1: female, 0: male	46.62%	-
age	in years	57.88	8.50
education	1: elementary school or above, 0: otherwise	64.17%	-
marital status	1: married with spouse present, 0: others	81.16%	-
retired	1: yes, 0: no	13.41%	-
income	in Yuan	10,783.51	11,056.88
**(2) Health status and functioning**			
number of conditions	(1) hypertension, (2) dyslipidemia, (3) diabetes or high blood sugar, (4) chronic lung disease, (5) heart problems, (6) kidney disease, (7) stomach or other digestive disease, (8) arthritis or rheumatism	1.1	1.17
pain	1: yes, 0: no	23.56%	-
BADL disability	(1) dressing, (2) bathing, (3) eating, (4) getting out of bed and walking across a room, (5) using the toilet and getting up and down, (6) controlling urination and defecation (1: had difficulty with at least one activity, 0: otherwise)	2.05%	-
**(3) Family and Social Connectivity**			
social engagement	(1) interacted with friends, (2) played ma-jong or cards or chess or went to community club, (3) went to a sport or social or other kind of club, (4) took part in a community-related organization, (5) took part in voluntary activity or charity, (6) attended an educational or training course (1: attained at least one activity, 0: otherwise)	51.31%	-
child interaction	Either live with all children or maintain at least weekly contact with non-coresident children (1: yes, 0: no)	65.13%	-
child support	1: yes, 0: no	31.24%	-

Mean: mean value of continuous variable; Prop: proportion of ones in binary variable; SD: standard deviation of continuous variable.

**Table 2 behavsci-15-01501-t002:** Multi-state model feature selection results.

	Est	Std Error	TRR	95% Lower CL	95% Upper CL	Sig
**(1) no symptom → new symptom episode**						
residence (urban)	−0.1660	0.0812	0.8471	0.7224	0.9932	*
sex (female)	0.4026	0.0568	1.4957	1.3382	1.6717	**
age	0.0000					
education (elementary or above)	0.0000					
marital status (married)	−0.0500	0.0683	0.9512	0.8321	1.0874	
retired (yes)	−0.2354	0.1050	0.7903	0.6432	0.9709	*
income	−0.6536	0.2866	0.5201	0.2966	0.9122	*
number of conditions	0.0000					
pain (yes)	0.3875	0.0606	1.4733	1.3083	1.6590	**
BADL disability (yes)	0.4671	0.1537	1.5954	1.1805	2.1561	*
social engagement (yes)	−0.1310	0.0543	0.8772	0.7886	0.9757	*
child interaction (yes)	0.0000					
child support (yes)	0.0000					
education × income	−0.5862	0.1144	0.5564	0.4446	0.6963	**
number of conditions × income	1.4512	0.3113	4.2681	2.3188	7.8560	**
number of conditions × sex	0.0000					
sex × age	0.0000					
age × education	0.0000					
**(2) new symptom episode → symptom persistence**						
residence (urban)	−0.0723	0.1490	0.9302	0.6946	1.2457	
sex (female)	0.0000					
age	−0.4992	0.3999	0.6070	0.2772	1.3292	
education (elemenatry or above)	0.0000					
marital status (married)	−0.0514	0.1238	0.9499	0.7453	1.2106	
retired (yes)	0.0000					
income	0.6114	0.5605	1.8430	0.6143	5.5291	
number of conditions	0.2197	0.7518	1.2457	0.2854	5.4364	
pain (yes)	0.0000					
BADL disability (yes)	0.0000					
social engagement (yes)	0.0000					
child interaction (yes)	0.0000					
child support (yes)	−0.0811	0.1157	0.9221	0.7350	1.1569	
education × income	−0.3622	0.2571	0.6962	0.4206	1.1523	
number of conditions × income	0.0000					
number of conditions × sex	−0.0814	0.5722	0.9218	0.3003	2.8294	
sex × age	0.5098	0.4215	1.6649	0.7288	3.8037	
age × education	0.0000					
**(3) new symptom episode/symptom persistence → symptom remission**						
residence (urban)	0.0000					
sex (female)	0.0000					
age	0.4673	0.2986	1.5956	0.8887	2.8648	
education (elemenatry or above)	0.0000					
marital status (married)	0.0821	0.1038	1.0855	0.8857	1.3303	
retired (yes)	0.0000					
income	−0.1472	0.7786	0.8631	0.1876	3.9703	
number of conditions	−0.1212	1.0128	0.8859	0.1217	6.4490	
pain (yes)	0.0000					
BADL disability (yes)	0.0000					
social engagement (yes)	0.0415	0.0811	1.0424	0.8892	1.2220	
child interaction (yes)	0.0000					
child support (yes)	0.0000					
education × income	0.1478	0.5211	1.1593	0.4175	3.2194	
number of conditions × income	0.0000					
number of conditions × sex	−0.2723	0.5577	0.7616	0.2553	2.2720	
sex × age	−0.3908	0.3525	0.6765	0.3390	1.3501	
age × education	0.0000					
**(4) symptom remission → symptom relapse**						
residence (urban)	−0.3314	0.3035	0.7179	0.3961	1.3014	
sex (female)	0.0000					
age	−1.0178	0.6924	0.3614	0.0930	1.4039	
education (elemenatry or above)	0.0000					
marital status (married)	0.0000					
retired (yes)	−0.4200	0.4482	0.6570	0.2729	1.5817	
income	0.0000					
number of conditions	0.0000					
pain (yes)	0.0000					
BADL disability (yes)	−0.2841	0.5022	0.7527	0.2813	2.0144	
social engagement (yes)	−0.0987	0.1900	0.9061	0.6244	1.3148	
child interaction (yes)	0.0000					
child support (yes)	0.0000					
education × income	0.0000					
number of conditions × income	0.0000					
number of conditions × sex	0.0000					
sex × age	1.5215	0.6106	4.5790	1.3835	15.1549	*
age × education	0.0000					

Est: estimated effect; Std Error: standard error of the estimated effect; TRR: transition rate ratio; 95% Lower CL: lower end of the 95% confidence limit of the transition rate ratio; 95% Upper CL: upper end of the 95% confidence limit of the transition rate ratio; Sig: significance of the risk factor, ** refers to *p*-value ≤ 0.001, * refers to *p*-value ≤ 0.05.

## Data Availability

This study utilized publicly accessible datasets, which are available at https://charls.charlsdata.com/pages/data/111/en.html.

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
