# Peer review of "Identifying Important Factors for Depressive Symptom Dynamics in Chinese Middle-Aged and Older Adults Using a Multi-State Transition Model with Feature Selection"

_behavsci, 2025, doi:10.3390/bs15111501_

Round 1
Reviewer 1 Report
Comments and Suggestions for Authors
The authors developed a dynamic multi-state model to explore the factors influencing the different stages of depressive symptoms. The findings are highly interesting and provide valuable insights; however, several points still need to be further consideration.
First, Although the authors accounted for the dynamic nature of depressive symptoms, they nevertheless adopted a CES-D cutoff of 12 points, effectively transforming the measure into a near-binary variable. Although this modeling strategy is appropriate and parsimonious, however, if the study aims to capture the longitudinal course of depression, further subdividing the “persistence” state into “worsening” and “stable,” and the “remission” state into “partial” and “full remission,” might provide a more nuanced understanding of symptom progression.
Second, the development and progression of depression are typically shaped by the interplay of psychological, social, and biological factors. Due to the limitations of the available database, not all potentially important variables—some of which may play crucial roles—were included in the model. And, the current framework seems to emphasize the independent effects of individual variables on depression transitions but does not explicitly consider potential interactions or synergistic effects among variables, which could represent a limitation.
Additionally, for the remission phase, it remains unclear whether participants received any form of treatment or pharmacotherapy, which may substantially influence the probability and pace of symptom improvement. As the study was conducted using a general older population cohort, this represents one of the inherent limitations of the database.
Overall, this study is methodologically innovative and offers meaningful implications for understanding and preventing depression, although the above considerations could further strengthen the interpretation of its findings.
Reviewer 2 Report
Comments and Suggestions for Authors
BEHAVIORAL SCIENCES
Comments for the Authors
Dear authors, I’ve read your paper with great interest since it is an original and valuable contribution to the knowledge of depressive disorders in a less explored segment of the population, which is steadily increasing. I commend you for the originality of the methodological proposal, which allows for examining the complexities of interactions between different factors and their changes over time through quantitative analysis. The writing of the article is clear and organized, helping the reader understand and follow this complex model of analysis. The discussion considers and comments on the association of each factor with depression through the four stages, as well as the results of other research on the subject. In order to contribute to the quality of the article, I’ll make just the following observations:
Discussion:
309: It says: elf-perceived health- quality of life , there is a missing letter? Should it be Self-perceived?
357-359: The Authors said that individuals who have never been married, particularly those in their middle and elderly years, often experience prolonged periods of loneliness, have less social support, and lower confidence. It will be convenient to cite the source of this statement.
Reviewer 3 Report
Comments and Suggestions for Authors
In the manuscript, the authors reported a study identifying contributing factors of difference levels of depressive symptoms from a national representative longitudinal dataset. The multi-state models were used to specify state transitions. The findings of the study have practical implications and will contribute to the literature. Please see detailed comments below.
- It is important to differentiate between clinical depression and elevated depressive symptoms in the introduction.
- Age of onset of the first major depressive episode is a critical factor. It would be helpful to include more literature of older age onset depression and studies compare depression of different onset ages
- What is the relationship between the assessments of the 10-item CES-D and clinical assessments of depression?
- An important feature of the study is to model the multiple state of depressive symptoms. Prior MDD episodes are the most effective predictors of next depressive episodes. Consider that, the State 3 and State 5 in Figure 2 might not be very different. It would be helpful to support the operational definitions of the states by evidence from clinical research.
